# SiameseNorm: Breaking the Barrier to Reconciling Pre/Post-Norm

Tianyu Li [* 1 2]   Dongchen Han [* 1]   Zixuan Cao [1]   Haofeng Huang [1]
Mengyu Zhou [2]   Ming Chen [2]   Erchao Zhao [2]   Xiaoxi Jiang [2]   Guanjun Jiang [2]   Gao Huang [1]

## Abstract

The long-standing tension between Pre- and Post-Norm remains an open problem in Transformer architecture, reflecting a fundamental trade-off between training stability and representational capacity. Prior attempts to combine their strengths have made progress, but often show limited robustness across training settings, restricting their broader applicability. We revisit this dilemma, showing that *single-stream* architectures struggle to reconcile Pre-Norm's stable identity-gradient propagation with Post-Norm's normalization of the main residual path. To address this structural tension, we propose SiameseNorm, a simple yet effective *two-stream* architecture that remains compatible with Pre-Norm training recipes. SiameseNorm couples Pre-Norm-like and Post-Norm-like streams through shared residual blocks, allowing each residual block to receive optimization signals from both pathways with negligible overhead. Extensive experiments on 400M and 1.3B dense language models, 15B MoE models, Vision Transformers, and Diffusion Transformers show that SiameseNorm consistently improves performance while maintaining strong training stability across architectures and modalities. Code is available at https://github.com/Qwen-Applications/SiameseNorm.

## 1. Introduction

The architectural evolution of deep neural networks has long reflected a trade-off between expressive representational capacity and training stability (Nair & Hinton, 2010; Krizhevsky et al., 2012; He et al., 2016; Ioffe & Szegedy, 2015; Huang et al., 2017). In Transformer archi-

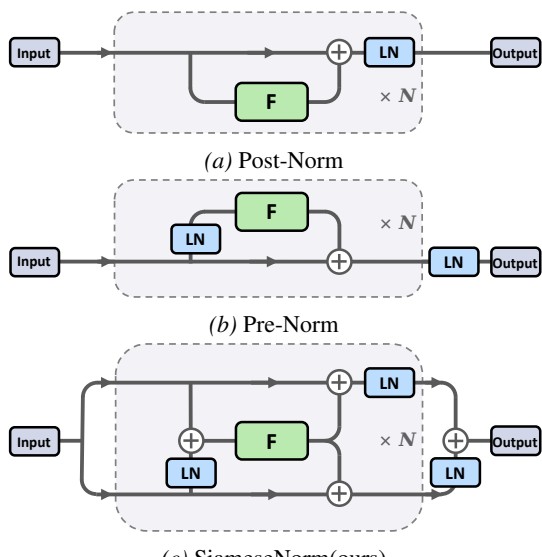

*(a)* Post-Norm

*(b)* Pre-Norm

*(c)* SiameseNorm(ours)

*Figure 1.* Architectural comparison of Post-Norm, Pre-Norm and SiameseNorm. In SiameseNorm, the input is duplicated into parallel streams sharing identical residual updates, where distinct LN positioning differentiates the hidden states across layers.

tectures (Vaswani et al., 2017), this tension is exemplified by the placement of Layer Normalization (LN) (Ba et al., 2016), a crucial module that facilitates optimization by standardizing hidden activations, yet simultaneously imposes strong rescaling on their magnitude. This architectural choice gives rise to two competing paradigms: Post-Norm (Figure 1a) places LN after residual addition, introducing stronger transformation dynamics along the main residual path but making training less stable; in contrast, Pre-Norm (Figure 1b) places LN only inside the residual branch, preserving an identity path that supports stable gradient propagation but weakens scale control over the main residual stream.

Despite the remarkable success of Pre-Norm in scaling modern Transformers to extremely large sizes (Brown et al., 2020; Touvron et al., 2023b; Liu et al., 2024; Yang et al., 2025; Dosovitskiy, 2020), the choice of normalization paradigm remains an open problem. In particular, recent studies observe that pruning a significant fraction of deep layers in Pre-Norm models often causes only negligible performance degradation (Gromov et al., 2025; Sun et al.,

*Equal contribution [1]Tsinghua University [2]Qwen Large Model Application Team, Alibaba. Correspondence to: Mengyu Zhou <zhoumengyu.zmy@alibaba-inc.com>, Gao Huang <gaohuang@tsinghua.edu.cn >.

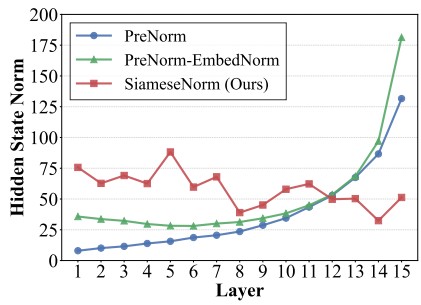

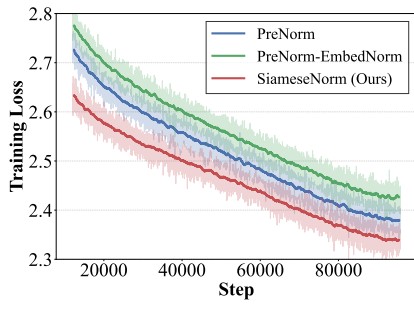

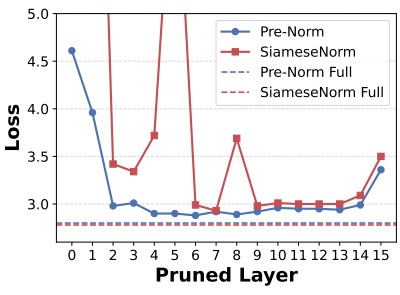

*(a)* Final layer-wise hidden-state magnitudes.

*(b)* Training loss curves.

*(c)* Loss changes after pruning individual layers.

*Figure 2.* Comparison of Pre-Norm, PreNorm with EmbedNorm, and our SiameseNorm using 1.3B models trained on 100B tokens with learning rate of $1 \times 10^{-3}$.

2025). This suggests that, although Pre-Norm provides excellent training stability, it may suffer from limited layer utilization and restricted effective depth.

On the other hand, Post-Norm is often observed to achieve stronger final performance when it can be stably optimized (Wang et al., 2024; Liu et al., 2020). However, as models scale up, ensuring stable and efficient training of Post-Norm Transformers becomes increasingly challenging. Moreover, empirical evidence (Liu et al., 2020; Xiong et al., 2020) shows that Post-Norm variants are highly sensitive to hyperparameters. As a result, directly applying standard Pre-Norm training recipes to Post-Norm architectures often leads to divergence or suboptimal performance, further limiting their practical adoption.

Extensive efforts in the community have been devoted to designing advanced normalization schemes with improved expressiveness, trainability, and scalability. A natural strategy is to combine Pre-Norm and Post-Norm to leverage the complementary strengths of both. In practice, however, such hybrid designs (Zhuo et al., 2025; Li et al., 2025; Wang et al., 2026) often lack training robustness outside specific settings, mirroring the instability of Post-Norm. We attribute this to a structural tension between the two paradigms: Pre-Norm stabilizes large-scale models by preserving an identity path that allows the signal magnitude to grow naturally, whereas Post-Norm restricts this growth by regulating the signal after each residual addition. Therefore, within a *single-stream* architecture, the desired properties of Pre-Norm and Post-Norm are difficult to reconcile.

Motivated by this insight, we propose **SiameseNorm**[1], a *two-stream* residual architecture that unifies Pre-Norm and Post-Norm. It maintains two coupled residual streams with shared computation modules: an unnormalized stream that

preserves the identity-gradient path of Pre-Norm, and a normalized stream that retains the representation dynamics of Post-Norm. By decoupling identity-gradient propagation from normalized representation learning and fusing the two streams by adding their normalized representations before each computation module, SiameseNorm combines the strengths of both paradigms with negligible overhead.

A compelling advantage of SiameseNorm is its compatibility with existing Transformer configurations. Adhering to the standard training recipes of Pre-Norm Transformers, we evaluate our method across a broad range of settings, including 400M and 1.3B dense language models, 15B MoE models, Vision Transformers, and Diffusion Transformers. Across model families and modalities, SiameseNorm consistently improves over Pre-Norm baselines while preserving optimization robustness. With only a simple architectural replacement and no architecture-specific tuning, SiameseNorm lowers the barrier to applying Post-Norm-style normalization and provides a practical foundation for further exploration.

## 2. Theoretical Motivation

We now formalize the behavior of Pre-Norm and Post-Norm residual blocks and analyze their gradient dynamics to highlight their respective optimization challenges, revealing the inherent structural tension of existing paradigms.

### 2.1. Notation

Consider a generic Transformer layer with input $X_i \in \mathbb{R}^d$ and output $X_{i+1} \in \mathbb{R}^d$, where $i = 0, \ldots, N-1$, and $N$ denotes the total number of layers. For any hidden state $X_i$, we define its **magnitude** as the $\ell_2$-norm, denoted by $\|X_i\|_2$. Throughout our analysis, we track the evolution of this magnitude across successive layers to characterize the signal scaling behavior of different normalization strategies.

Let $F_i(\cdot)$ denote the residual transformation (e.g., Attention

---

[1]This name is inspired by Siamese networks (Bromley et al., 1993). It reflects our architecture's use of twin, coupled residual streams processing distinct normalization schemes through shared computation modules.

or MLP) and $\theta_i$ denote its parameters, so that $\nabla_{\theta_i}\mathcal{L}$ represents the parameter gradients. We use LN to denote generic normalization (e.g., LayerNorm or RMSNorm), since the specific variant does not alter the qualitative gradient dynamics discussed herein. Finally, let $\mathbf{J}_F \triangleq \frac{\partial F(X)}{\partial X}$ denote the Jacobian of a function $F$.

## 2.2. Evaluation of Existing Paradigms

We categorize existing architectures based on the evolution of their hidden state magnitudes.

**Pre-Norm** Pre-Norm paradigm(Figure 1b) places LN only inside the residual branch:

$$X_{i+1} = X_i + F_i(\mathrm{LN}_i(X_i)) \qquad (1)$$

This family includes standard Pre-Norm and variants like Peri-LN (Kim et al., 2025). The loss gradient $\mathcal{L}$ with respect to $\theta_i$ in Equation (1) is given by:

$$\nabla_{\theta_i}\mathcal{L} = \frac{\partial\mathcal{L}}{\partial X_N}\left(\prod_{j=N-1}^{i+1}\frac{\partial X_{j+1}}{\partial X_j}\right)\frac{\partial X_{i+1}}{\partial \theta_i} \qquad (2)$$

$$= \frac{\partial\mathcal{L}}{\partial X_N}\left[\prod_{j=N-1}^{i+1}\left(\mathbf{I} + \frac{\partial F_j\left(\mathrm{LN}_j(X_j)\right)}{\partial X_j}\right)\right]\frac{\partial X_{i+1}}{\partial\theta_i} \qquad (3)$$

$$= \frac{\partial\mathcal{L}}{\partial X_N}\left[\prod_{j=N-1}^{i+1}\left(\mathbf{I} + \mathbf{J}_{F_j}\mathbf{J}_{\mathrm{LN}_j}\right)\right]\frac{\partial X_{i+1}}{\partial\theta_i}, \qquad (4)$$

where the product denotes an ordered composition of Jacobians from layer $N-1$ down to $i+1$. The identity term $\mathbf{I}$ induced by the skip connection provides an explicit gradient highway, which is a key reason for the optimization stability of Pre-Norm. However, it allows the representation magnitudes to grow unbounded (Kim et al., 2025), as shown in Figure 2a. This creates a scale mismatch in deeper layers: each block receives normalized inputs but must update an increasingly large main path. As a result, the relative contribution of deeper blocks is diluted, limiting the effective depth of Pre-Norm.

To test this intuition, we apply a parameter-free RMSNorm after the embedding layer, yielding PreNorm-EmbedNorm in Figure 2a. This rescales the initial hidden magnitude from approximately 2 to $\sqrt{d}$, about 45 in our setup. Although it flattens the early-layer magnitude profile, it worsens perplexity by 0.4, suggesting that magnitude regulation in Pre-Norm is challenging.

**Post-Norm** Post-Norm paradigm(Figure 1a) applies LN after the residual addition:

$$X_{i+1} = \mathrm{LN}_i(X_i + F_i(X_i)) \qquad (5)$$

This family includes standard Post-Norm and variants such as DeepNorm (Wang et al., 2024), HybridNorm (Zhuo et al., 2025), and SpanNorm (Wang et al., 2026). By placing normalization on the main path, Post-Norm keeps the hidden representation scale regulated across depth, allowing each block to exert a stronger influence than in Pre-Norm. However, this benefit is accompanied by optimization instability, since backpropagation requires multiplying gradients by the LN Jacobian at each layer:

$$\nabla_{\theta_i}\mathcal{L} = \frac{\partial\mathcal{L}}{\partial X_N}\left(\prod_{j=N-1}^{i+1}\frac{\partial X_{j+1}}{\partial X_j}\right)\frac{\partial X_{i+1}}{\partial\theta_i} \qquad (6)$$

$$= \frac{\partial\mathcal{L}}{\partial X_N}\left[\prod_{j=N-1}^{i+1}\mathbf{J}_{\mathrm{LN}_j}\left(\mathbf{I} + \mathbf{J}_{F_j}\right)\right]\frac{\partial X_{i+1}}{\partial\theta_i} \qquad (7)$$

Even if $\mathbf{I} + \mathbf{J}_{F_j}$ is well-conditioned, repeated application of $\mathbf{J}_{\mathrm{LN}}$ causes multiplicative instability. Since the spectral norm of $\mathbf{J}_{\mathrm{LN}}$ is sensitive to the signal, this compounding effect can cause gradients to vanish or explode as the depth $N$ increases. This mechanism explains the severe optimization instability observed in deep Post-Norm Transformers.

## 2.3. Structural Tension

Although prior approaches (Li et al., 2025; Zhuo et al., 2025; Wang et al., 2026) attempt to combine the benefits of Pre-Norm and Post-Norm, our empirical observations in Table 1 suggest that they often remain constrained by the trade-off between stability and performance. We argue that this limitation is not merely an implementation detail, but reflects a structural tension between the two paradigms.

**The Dilution Problem in Pre-Norm.** By maintaining a clean identity path, Pre-Norm supports stable gradient propagation. However, this comes at the cost of unbounded magnitude growth. As illustrated in Figure 2a, while the input to the residual block is constrained to a constant level, the hidden state magnitude exhibits near-exponential growth. This growing disparity creates a severe optimization burden: to maintain a meaningful relative contribution against the massive main path, deeper layers are compelled to learn increasingly large output magnitudes. The difficulty of learning such extreme scalings typically results in signal dilution, severely limiting the model's effective depth.

**The Distortion Problem in Post-Norm.** By periodically bounding hidden state magnitudes, Post-Norm aims to unlock higher layer utilization. However, this scale control comes at the cost of repeated shrinkage. After each residual addition, the accumulated representation is rescaled back to a fixed range, which can distort the signal. These repeated rescalings introduce normalization Jacobians along the gra-

**Algorithm 1** SiameseNorm forward pass

1: $h \leftarrow \text{Embed}(x)$
2: $X_0 \leftarrow h, Y_0 \leftarrow h$
3: **for** $i = 0, \ldots, N-1$ **do**
4:     $O \leftarrow F_i\big(X_i + \text{LN}_i^Y(Y_i)\big)$
5:     $X_{i+1} \leftarrow \text{LN}_i^X(X_i + O)$
6:     $Y_{i+1} \leftarrow Y_i + O$
7: **end for**
8: **return** $X_N + \text{LN}_{\text{final}}(Y_N)$

dient path, making optimization more sensitive and often less stable.

Existing hybrid methods partially alleviate this tension by assigning Pre-Norm and Post-Norm behaviors to different layers or submodules. However, because all updates are still accumulated along a single shared main path, the same representation must simultaneously support two conflicting roles: preserving an unnormalized identity-gradient route for stable optimization, and applying repeated normalization to control residual-state magnitudes. These two requirements are difficult to satisfy within one stream. Therefore, rather than directly mixing Pre-Norm and Post-Norm operations inside a single residual path, we are motivated to structurally decouple their roles into separate but coupled streams.

## 3. SiameseNorm

We now introduce **SiameseNorm**. As illustrated in Figure 1c, SiameseNorm maintains two coupled residual streams, denoted as $X_i$ and $Y_i$. The $X_i$ stream is normalized after each residual update, resembling a Post-Norm path that keeps the hidden-state magnitude controlled. The $Y_i$ stream accumulates residual updates without normalization on the main path, resembling a Pre-Norm path that preserves an identity route for stable gradient propagation. At each layer, both streams interact through a shared residual block $F_i$, so the architecture introduces a negligible parameter overhead. The forward pass is summarized in Algorithm 1.

**Generalization Capabilities**   SiameseNorm connects multiple normalization paradigms through simple parameter configurations. Zeroing $\text{LN}_i^X$ recovers a Pre-Norm topology, while zeroing $\text{LN}_i^Y$ yields a Post-Norm-style architecture. Layer-wise stream selection further captures hybrid switching schemes such as Mix-LN (Li et al., 2025), where earlier layers use Post-Norm and later layers use Pre-Norm. Overall, SiameseNorm spans a broader design space of Pre-Norm-like, Post-Norm-like, and hybrid-like behaviors.

**Gradient Analysis**   Let $S_i = [X_i, Y_i]^\top$ be the concatenated state of the two streams. Examine the loss gradient $\mathcal{L}$

with respect to the residual transformation parameters $\theta_i$:

$$\nabla_{\theta_i}\mathcal{L} = \frac{\partial\mathcal{L}}{\partial S_N}\left(\prod_{j=N-1}^{i+1}\frac{\partial S_{j+1}}{\partial S_j}\right)\frac{\partial S_{i+1}}{\partial O_i}\frac{\partial O_i}{\partial\theta_i} \quad (8)$$

$$= \frac{\partial\mathcal{L}}{\partial S_N}\left(\prod_{j=N-1}^{i+1}\frac{\partial S_{j+1}}{\partial S_j}\right)\begin{bmatrix}\mathbf{J}_{\text{LN}_i^X}\\\mathbf{I}\end{bmatrix}\frac{\partial O_i}{\partial\theta_i} \quad (9)$$

The block Jacobian transition matrix is given by:

$$\frac{\partial S_{j+1}}{\partial S_j} = \begin{bmatrix}\mathbf{J}_{\text{LN}_j^X}(\mathbf{I} + \mathbf{J}_{F_j}) & \mathbf{J}_{\text{LN}_j^X}\mathbf{J}_{F_j}\mathbf{J}_{\text{LN}_j^Y}\\\mathbf{J}_{F_j} & \mathbf{I} + \mathbf{J}_{F_j}\mathbf{J}_{\text{LN}_j^Y}\end{bmatrix} \quad (10)$$

The diagonal blocks of this transition matrix reveal structural connections to both standard normalization paradigms. The bottom-right block, $\mathbf{I} + \mathbf{J}_{F_j}\mathbf{J}_{\text{LN}_j^Y}$, matches the Pre-Norm transition in Equation (4), providing a direct identity-gradient route through the $Y$-stream. The top-left block, $\mathbf{J}_{\text{LN}_j^X}(\mathbf{I}+\mathbf{J}_{F_j})$, resembles the Post-Norm transition in Equation (7), yielding a normalized residual path through the $X$-stream. Together with the off-diagonal coupling terms, SiameseNorm separates Pre-Norm-like gradient propagation and Post-Norm-like residual normalization into two interacting streams. Since both streams share the same residual update $O_i$, each residual block can receive optimization signals from both pathways with negligible overhead.

**Auxiliary Mechanisms**   To make SiameseNorm compatible with existing training recipes, we introduce two auxiliary mechanisms: **Normalized Input** and **Depth-wise Scaling**.

First, we apply an extra LN to the aggregated representation before the shared residual block, ensuring a stable input distribution consistent with standard Transformers.

Second, motivated by DeepNorm (Wang et al., 2024), we scale the residual update sent to the bounded Post-Norm stream by $1/\sqrt{l+1}$, where $l$ denotes the layer index. From the optimization perspective, this depth-dependent scaling reduces the sensitivity of the Post-Norm-style pathway and better aligns its initial update magnitude with the stable Pre-Norm training recipe, enabling the use of higher learning rates. Furthermore, as depth increases, a scale mismatch naturally arises between the two streams: the hidden-state norms in the Pre-Norm stream tend to grow, whereas the Post-Norm stream remains bounded by normalization. This creates a dilemma in deep layers: the shared residual update may be too small to meaningfully affect the growing Pre-Norm stream, yet too large to maintain stability in the bounded stream. Depth-wise Scaling mitigates this mismatch by attenuating the update injected into the bounded stream, thereby rebalancing the contributions of the two pathways while preserving training stability.

*Table 1.* Evaluation results on dense 1.3B models and 15A2B MoE models. We report perplexity (PPL) and accuracy on downstream tasks across three learning rate settings ($4 \times 10^{-4}$, $1 \times 10^{-3}$ and $2 \times 10^{-3}$). Entries marked as *diverge* denote cases where the model failed to converge, and entries marked with $^*$ indicate training runs with loss spikes, signifying training instability.

| Method | Avg. PPL↓ | ARC-E | ARC-C | HellaSwag | OpenBookQA | PIQA | WinoGrande | Arith. | Avg. Score ↑ |
|---|---|---|---|---|---|---|---|---|---|
| *Setting A: Conservative Learning Rate* ($\eta = 4 \times 10^{-4}$), **100B tokens** | | | | | | | | | |
| Post-Norm | 12.61 | 67.5 | 34.1 | 48.0 | 36.6 | 70.6 | 53.6 | 27.1 | 48.21 |
| Deep-Norm | 12.95 | 64.9 | 30.1 | 46.5 | 36.4 | 69.2 | 55.7 | 26.3 | 47.01 |
| ResiDual | 12.32 | 66.8 | 33.4 | 49.6 | 36.8 | 71.1 | 54.5 | 25.8 | 48.29 |
| Pre-Norm | 11.21 | 70.2 | 37.5 | 54.1 | 36.2 | 73.3 | 56.5 | 27.7 | 50.79 |
| Peri-LN | 11.25 | 67.0 | 37.5 | 53.9 | 38.4 | 72.8 | 55.8 | 26.7 | 50.30 |
| Hyper-Connections-2×DHC | 11.12 | 70.9 | 38.8 | 54.6 | 39.0 | 72.6 | 55.2 | 26.9 | 51.14 |
| SpanNorm | 11.00 | 69.6 | 38.5 | 56.1 | **42.6** | 73.9 | 55.8 | 28.1 | 52.09 |
| HybridNorm | 10.91 | 72.3 | 38.8 | 56.4 | 38.4 | 74.2 | 57.0 | 27.6 | 52.10 |
| **SiameseNorm(Ours)** | **10.57** | 72.1 | 36.8 | 57.8 | 40.6 | 72.5 | 57.6 | 28.4 | **52.26** |
| *Setting B: High Learning Rate* ($\eta = 1 \times 10^{-3}$), **100B tokens** | | | | | | | | | |
| Post-Norm | *diverge* | - | - | - | - | - | - | - | - |
| Deep-Norm | 11.47 | 68.4 | 35.4 | 53.7 | 38.8 | 72.1 | 56.2 | 27.7 | 50.33 |
| ResiDual | 11.22$^*$ | 70.0 | 39.5 | 55.1 | 41.2 | 72.8 | 57.0 | 27.5 | 51.87 |
| Pre-Norm | 10.84 | 71.9 | 37.8 | 56.4 | 39.8 | 73.8 | 56.7 | 27.0 | 51.91 |
| Peri-LN | 10.84 | 70.5 | 38.1 | 56.2 | 39.4 | 73.7 | 57.3 | 28.5 | 51.96 |
| Hyper-Connections-2×DHC | 10.73 | 72.6 | 39.1 | 57.7 | 41.2 | 73.8 | 59.9 | 27.9 | 53.17 |
| SpanNorm | 10.86 | 71.9 | 37.5 | 56.7 | 38.4 | 74.1 | 58.2 | 28.6 | 52.20 |
| HybridNorm | *diverge* | - | - | - | - | - | - | - | - |
| **SiameseNorm (Ours)** | **10.43** | 72.5 | 39.8 | 59.0 | 41.0 | 74.0 | 59.0 | 29.4 | **53.53** |
| *Setting C: Aggressive Learning Rate* ($\eta = 2 \times 10^{-3}$), **100B tokens** | | | | | | | | | |
| Deep-Norm | *diverge* | - | - | - | - | - | - | - | - |
| HybridNorm | *diverge* | - | - | - | - | - | - | - | - |
| ResiDual | 13.66$^*$ | 64.4 | 33.8 | 45.2 | 35.4 | 69.0 | 53.6 | 26.2 | 46.80 |
| Pre-Norm | 10.89 | 71.6 | **40.8** | 57.7 | 41.4 | 73.7 | 57.3 | 28.1 | 52.94 |
| Peri-LN | 10.89 | 71.4 | 40.1 | 57.8 | 41.2 | 73.8 | 59.7 | 28.7 | 53.24 |
| Hyper-Connections-2×DHC | 10.77 | **73.7** | 40.1 | 58.2 | 40.0 | 73.8 | 60.4 | 30.6 | 53.83 |
| SpanNorm | diverge | - | - | - | - | - | - | - | - |
| **SiameseNorm (Ours)** | **10.48** | 73.5 | 38.5 | **59.6** | 41.4 | **74.6** | **62.2** | **39.6** | **55.63** |
| *Setting D: Aggressive Learning Rate* ($\eta = 2 \times 10^{-3}$), **350B tokens** | | | | | | | | | |
| Pre-Norm | 9.67 | **78.1** | 42.8 | 63.5 | 41.6 | **76.0** | 62.0 | 36.2 | 57.17 |
| Hyper-Connections-2×DHC | 9.57$^*$ | 76.3 | 43.1 | 63.5 | 42.4 | 75.3 | 61.3 | 33.6 | 56.50 |
| **SiameseNorm (Ours)** | **9.42** | 75.6 | **44.1** | 64.7 | **44.6** | 76.0 | 62.5 | 43.4 | **58.70** |
| *Setting E: 15A2B MoE* ($\eta = 1 \times 10^{-3}$), **100B tokens** | | | | | | | | | |
| Pre-Norm | 7.92 | **78.9** | 47.8 | 69.4 | **45.2** | 77.1 | 61.8 | 48.6 | 61.26 |
| **SiameseNorm (Ours)** | **7.76** | 78.9 | 48.8 | 69.9 | 44.0 | **77.9** | 63.8 | 58.2 | **63.07** |

Importantly, these practical modifications only affect the internal definition and scale of the residual Jacobian $\mathbf{J}_{F_j}$, while preserving the two-stream transition structure. Therefore, the structural conclusions above remain unchanged. We empirically analyze their contributions in Section 4.5.

**Computational Overhead** SiameseNorm introduces only auxiliary LN operations, yielding negligible overhead compared to standard Pre-Norm Transformers. Because LN is lightweight relative to the dominant Attention and MLP blocks in both parameters and computation, the additional normalization operations have only a marginal impact on the overall complexity. Theoretically, the parameter count and FLOPs increase by under 0.1%. Empirically, scaling to a 15B MoE model results in only a 0.5% decrease in training speed and a 2% increase in activation memory.

## 4. Experiments

We first compare SiameseNorm with existing normalization paradigms in large-scale language-model pretraining. Beyond this main setting, we further conduct experiments across depth and modality. Specifically, we perform fixed-parameter depth-width studies and test SiameseNorm on Vision Transformers and Diffusion Transformers.

### 4.1. Language Experimental Setup

We conduct controlled comparisons based on the OLMo (Groeneveld et al., 2024) architecture trained from scratch on FineWeb-Edu (Penedo et al., 2024). We compare SiameseNorm with representative normalization strategies, including Pre-Norm, Peri-LN (Kim et al., 2025), Post-Norm, DeepNorm (Wang et al., 2024), ResiDual (Xie et al., 2023),

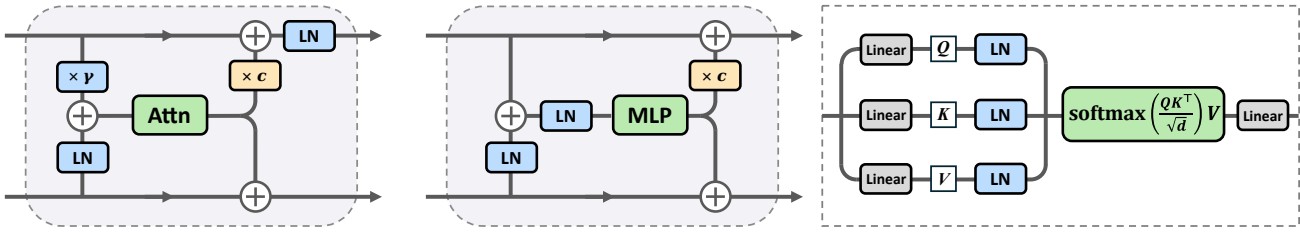

*(a)* Macro view of the Attention Layer.     *(b)* Macro view of the MLP layer.     *(c)* Micro view of HybridNorm Attention block.

*Figure 3.* In Language pretraining setting, SiameseNorm couples HybridNorm (Zhuo et al., 2025) and Pre-Norm with HybridNorm Attention blocks.

HybridNorm (Zhuo et al., 2025), SpanNorm (Wang et al., 2026) and Hyper-Connections-2×DHC (Zhu et al., 2025a). To compare different architectures across learning-rate settings, we train different architectures with learning rates of $4 \times 10^{-4}$, $10^{-3}$, and $2 \times 10^{-3}$ for 100B tokens, and further extend the aggressive $2 \times 10^{-3}$ setting to 350B tokens to assess long-term stability. Finally, we compare Pre-Norm and SiameseNorm on a large-scale MoE setting with 15B total parameters and 2B active parameters, based on OL-MoE (Muennighoff et al., 2025). The total computational cost exceeds 60,000 A100 hours. Detailed settings are provided in Section A.4.

Models are evaluated across benchmarks including ARC (Clark et al., 2018), HellaSwag (Zellers et al., 2019), PIQA (Bisk et al., 2020), WinoGrande (Sakaguchi et al., 2021), OpenBookQA (Mihaylov et al., 2018) and Arithmetic (Brown et al., 2020). We also report average perplexity (PPL) on the training set.

In practice, SiameseNorm combines a Pre-Norm stream with a HybridNorm-style stream (Zhuo et al., 2025), using the latter as a competitive Post-Norm-style baseline. Detailed implementations are provided in Figure 3. Unlike previous multi-path methods (Zhu et al., 2025a; Xie et al., 2025) that rely on Pre-Norm-biased initialization, we initialize all LN scales to 1.0, enforcing equal initial stream contribution and directly testing the intrinsic stability of SiameseNorm.

### 4.2. Main Results

Table 1 summarizes the performance of various normalization architectures across different learning rate regimes. Our empirical findings lead to the following observations:

**Learning Rate Sensitivity Obscures Architectural Comparisons** Under conservative learning rates ($\eta = 4 \times 10^{-4}$), all methods converge successfully, with HybridNorm and SpanNorm outperforming Pre-Norm, illustrating the representational potential of the Post-Norm paradigm. However, scaling the learning rate to $\eta = 1 \times 10^{-3}$ explicitly exposes the underlying instability of Post-Norm architec-

tures, causing both standard Post-Norm and the previously competitive HybridNorm to diverge. Pushing the learning rate further to $\eta = 2 \times 10^{-3}$ dramatically widens this stability gap: all evaluated Post-Norm-style variants either diverge or exhibit severe loss spikes. In stark contrast, Pre-Norm-style architectures, including standard Pre-Norm and Peri-LN, maintain robust convergence under these aggressive settings. Notably, their downstream task performance continues to scale with the learning rate, indicating that their structural stability enables them to fully leverage faster optimization regimes. Consequently, learning-rate sensitivity fundamentally complicates architectural comparisons: while Post-Norm variants appear competitive under carefully tuned conservative environments, their severe optimization instability ultimately prevents them from capitalizing on the performance gains offered by larger learning rates.

**Superiority of SiameseNorm** Across all learning rate configurations, our proposed SiameseNorm demonstrates exceptional training stability and performance. In particular, it achieves the best PPL of $10.43$ in Setting B, which represents a significant **reduction of 0.3** compared to the strongest baseline, underscoring that our fusion approach not only inherits the optimization benefits of Pre-Norm but also leverages the superior expressive capacity of Post-Norm architectures. Furthermore, with a higher learning rate of $2 \times 10^{-3}$, SiameseNorm achieves a substantial accuracy of $39.6\%$ on Arithmetic tasks, far exceeding the random baseline $25\%$ and the range $28\%$ to $31\%$ typical of other methods. This **41% relative improvement** over Pre-Norm further validates the superior capacity of our architecture for sequential reasoning.

### 4.3. Generality Across Depths and Modalities

We further evaluate the generality of SiameseNorm along two axes: robustness across model shapes and transfer beyond language modeling to image classification and image generation.

For language modeling, we train approximately 390M-parameter models with different depth-width configurations for 12B tokens using a learning rate of $1 \times 10^{-3}$. As shown

*Table 2.* Generality results across model depths and beyond language modeling. Top: language modeling experiments with varying depths under a fixed training budget. Bottom: cross-modal results on image classification and image generation.

| Depth Generality in Language Modeling | | | |
|---|---|---|---|
| Layers/Dim | Pre-Norm | SiameseNorm | PPL Reduction |
| 10/1280 | 17.47 | 16.15 | 1.32 |
| 17/1024 | 17.23 | 15.69 | 1.54 |
| 33/768 | 17.29 | **15.64** | 1.65 |
| 80/512 | 18.02 | 15.98 | **2.04** |
| Cross-Modal Generality | | | |
| Model | Metric | Pre-Norm | SiameseNorm |
| DeiT-T ($L = 12$) | Acc ↑ | 72.2 | **73.6** |
| DeiT-S ($L = 12$) | Acc ↑ | 79.8 | **81.3** |
| DiT-B/2 ($L = 12$) | FID ↓ | 42.43 | **40.31** |
| DiT-L/4 ($L = 24$) | FID ↓ | 45.21 | **41.34** |

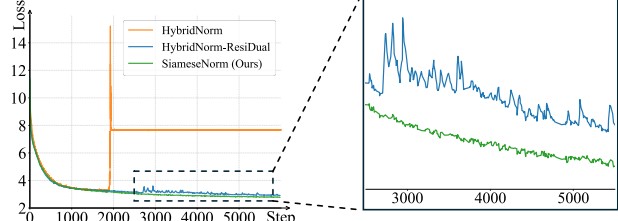

*Figure 4.* Training loss curves of HybridNorm (yellow), HybridNorm with ResiDual (blue) and our SiameseNorm (green) without Depth-wise Scaling.

in Table 2, SiameseNorm consistently outperforms Pre-Norm across all configurations. Moreover, the improvement becomes larger in deeper models: the largest perplexity reduction appears at 80 layers, and SiameseNorm achieves its best perplexity at 33 layers, whereas Pre-Norm performs best at 17 layers and slightly degrades at 33 layers. These results suggest that SiameseNorm better utilizes increased depth under a fixed parameter budget.

Beyond language modeling, we apply the same Pre-Norm + Post-Norm Siamese construction equipped with Normalized Input and Depth-wise Scaling to DeiT (Touvron et al., 2021a) and DiT (Peebles & Xie, 2023) under standard ImageNet (Deng et al., 2009) training and evaluation settings, without additional tuning. The consistent improvements in Top-1 accuracy and FID show that SiameseNorm can serve as a drop-in architectural modification across different Transformer families. In particular, the larger gain on DiT-L/4 (24 layers) than on DiT-B/2 (12 layers) further supports the view that SiameseNorm improves the propagation and integration of layer-wise transformations in deeper models.

### 4.4. Ablation Study of Siamese Topology

The Siamese topology is the central architectural contribution of our method. To isolate its effect, we compare three topologies built upon the HybridNorm setting: the original single-stream topology, ResiDual (Xie et al., 2023), and our Siamese design without Depth-wise Scaling. As shown in Figure 4 and Rows 1, 3 and 5 in Table 3, the baselines either diverge or suffer from degraded performance under the aggressive learning rate of $10^{-3}$. In contrast, the Siamese topology achieves smooth convergence and reaches a PPL of 10.68, outperforming both the diverging HybridNorm baseline and the standard Pre-Norm baseline (10.84). Notably, even under the conservative matched comparison with both auxiliary mechanisms enabled, replacing the original topology with the Siamese topology reduces perplexity by

0.22 (Rows 2 vs. 7), which is already a non-trivial margin in large-scale language model pretraining (Qiu et al., 2025; Xie et al., 2025; Team et al., 2026).

Moreover, warm-up stress tests further highlight the robustness of the Siamese topology. Prior studies (Xiong et al., 2020; Nguyen & Salazar, 2019a) have identified tolerance to short or even absent warm-up schedules as an important empirical indicator of Pre-Norm stability, whereas Post-Norm variants are typically much more sensitive to warm-up. Under the standard 2K-step warm-up, SiameseNorm without Depth-wise Scaling achieves a perplexity comparable to HybridNorm with Depth-wise Scaling (10.68 vs. 10.65). However, under a more challenging shortened warm-up setting, HybridNorm already diverges at **300** warm-up steps, whereas SiameseNorm remains stable even **without warm-up**. This contrast suggests that, beyond the effect of residual scaling, the Siamese topology inherits Pre-Norm-like robustness while retaining the performance potential of the Post-Norm-style stream.

### 4.5. Ablation Study of Auxiliary Mechanisms

**Effect of Depth-wise Scaling** Although Depth-wise Scaling is an established technique (Wang et al., 2024; Sun et al., 2025; Chen & Wei, 2026), Table 3 demonstrates its vital necessity within our framework. Following the theoretical insights from Wang et al. (2024), this scaling mechanism reduces the magnitude of model updates, effectively aligning the initial optimization dynamics of the residual branches with the Pre-Norm recipe. Empirically, it not only elevates the maximum stable learning rate to prevent divergence in baseline Post-Norm architectures, but also synergizes seamlessly with SiameseNorm.

**Necessity of Normalized Input** Providing normalized input to the Attention and MLP modules is a universal and critical characteristic of modern Transformer training. While the sub-streams are individually normalized prior to fusion, our ablation (Row 4 vs. Row 5 and Row 6 vs. Row 7 in Table 3) confirms that normalizing the fused representation is indispensable. This performance gain underscores that preserving this standard normalization property at the

*Table 3.* Ablation study of key components in SiameseNorm with learning rate of $10^{-3}$. The first and last rows denote HybridNorm and our SiameseNorm respectively. Note that normalized input is inherent to HybridNorm and standard Pre/Post-Norm architectures.

| Normalized Input | Depth-Scaling | Topology | Avg. PPL ↓ |
|:---:|:---:|:---:|:---:|
| ✓ | × | Original | *diverge* |
| ✓ | ✓ | Original | 10.65 |
| ✓ | × | ResiDual | 11.68* |
| × | × | Siamese | 10.88 |
| ✓ | × | Siamese | 10.68 |
| × | ✓ | Siamese | 10.51 |
| ✓ | ✓ | Siamese | **10.43** |

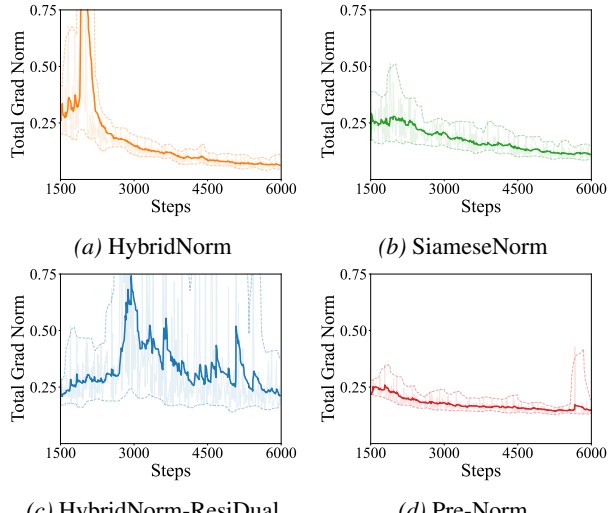

*(a)* HybridNorm      *(b)* SiameseNorm

*(c)* HybridNorm-ResiDual      *(d)* Pre-Norm

*Figure 5.* Gradient norm comparisons

block input level is fundamental for efficient optimization.

In summary, Depth-wise Scaling and Normalized Input are not our innovations, but lightweight auxiliary components that make SiameseNorm compatible under existing Pre-Norm training recipes.

## 5. Analysis

We now analyze why SiameseNorm works, focusing on gradient statistics and LN parameters.

### 5.1. Optimization Stability

As shown in Figure 5, to investigate the training stability under high learning rates of $1 \times 10^{-3}$, we monitor the gradient norms of different architectures throughout training. As hypothesized, the Post-Norm variant HybridNorm exhibits extreme instability. We observe severe gradient explosions, with gradient norms repeatedly spiking to magnitudes exceeding 100. Such oscillations typically lead to irreversible training divergence. In sharp contrast, SiameseNorm maintains a stable optimization trajectory comparable to Pre-Norm: After the warm-up phase, the gradient norms for both SiameseNorm and Pre-Norm consistently remain below 0.5. This confirms that SiameseNorm successfully inherits the optimization stability of Pre-Norm, effectively mitigating the gradient explosion issues of Post-Norm architectures and enabling more aggressive learning rates.

### 5.2. Layer Utilization and Effective Depth

To evaluate whether SiameseNorm improves layer utilization, we remove individual Transformer layers from trained models and measure the resulting loss increase on the C4 (Raffel et al., 2020) validation set. A larger degradation indicates a stronger contribution from the pruned layer.

As shown in Figure 2c, SiameseNorm consistently exhibits larger loss increases than the Pre-Norm baseline across pruned layers, suggesting that its layers are more actively

utilized and make stronger contributions to model performance. This is consistent with the hidden-state magnitude analysis in Figure 2a: by periodically normalizing the main hidden state, SiameseNorm keeps the X-stream magnitude stable across depth, which can improve the effectiveness of deep residual updates. The effect is also pronounced on the arithmetic task, which requires stronger sequential reasoning. Together, these results provide evidence that SiameseNorm better exploits deeper layer transformations and alleviates the effective-depth limitation of Pre-Norm architectures.

### 5.3. Contribution of Each Stream

We further investigate the internal dynamics of SiameseNorm by analyzing the mixing intensity of each stream to the input and conducting a Logit Lens analysis.

**Input Intensity Comparison** To investigate the internal dynamics of SiameseNorm across layers, we analyze the mixing proportion of each stream to the input. Specifically, we extract the learned scaling parameters of LN from both the Hybrid-Norm stream (X-stream) and the Pre-Norm stream (Y-stream) and normalize them to derive their relative contribution ratios. As illustrated in Figure 6, we observe that for the vast majority of residual blocks, both streams maintain significant proportions. This indicates that SiameseNorm effectively leverages hidden representations from both streams, ensuring that they jointly contribute to the feature extraction process.

**Post-Norm Variant Stream Dominance** We examine the average learned LN weights at the final fusion layer to quantify each stream's contribution to the output. The HybridNorm stream converges to a significantly larger weight

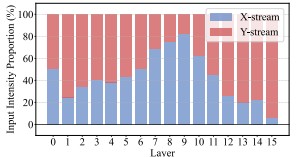
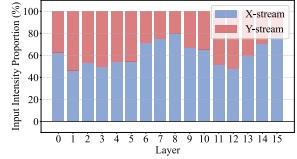

*(a) Attention blocks*       *(b) MLP blocks*

*Figure 6.* Comparison of scales ratios for input between the Hybrid-Norm stream (blue) and the Pre-Norm stream (red).

(1.05) compared to the Pre-Norm stream (0.42). Furthermore, drawing on the intuition of Logit Lens (Geva et al., 2021), we project the final hidden states of each stream directly into the vocabulary space to identify which stream drives model decisions. The HybridNorm stream exhibits clear dominance, matching the final output 42.6% of the time, compared to only 16.2% for the Pre-Norm stream. In divergent predictions, the model aligns with the Hybrid-Norm stream 41.2% of the time, versus just 14.3% for Pre-Norm. This dominance confirms that our approach successfully unlocks the potential of the Post-Norm paradigm.

## 6. Related Work

**Macro-Architectural Designs**  Foundational residual architectures, such as ResNet (He et al., 2016), Highway Networks (Srivastava et al., 2015) and DenseNet (Huang et al., 2017), demonstrated that explicit shortcut or gating pathways substantially ease optimization and improve depth scalability. A substantial body of work (Wang et al., 2024; Touvron et al., 2021b; Bachlechner et al., 2021) improves very-deep optimization through initialization and residual scaling in Transformer. Another line of work modifies the residual topology to alleviate the trade-off between gradient stability and depth utilization: ResiDual (Xie et al., 2023) mitigates Post-Norm instability via per-block shortcuts to the output, while hybrid strategies like Mix-LN (Li et al., 2025), HybridNorm (Zhuo et al., 2025) and Span-Norm (Wang et al., 2026) combine Pre-Norm and Post-Norm across layers to balance signal propagation. Recent hidden dimension scaling schemes (Baykal et al., 2023; Zhu et al., 2025a;b; Xie et al., 2025; Han et al., 2025) leverage adaptive widening connections, yet remain confined to the Pre-Norm paradigm. We provide a comprehensive discussion and distinction between our approach and these topology-modifying works in supplementary material.

**Micro-Architectural Designs**  Complementary to macro-topological changes, substantial progress has been made in optimizing the internal sub-layers of the Transformer. In the Feed-Forward Network (FFN), activation functions have evolved from GeLU (Hendrycks, 2016) to GLU variants, particularly SwiGLU (Shazeer, 2020), which have become the default in modern LLMs for their superior performance. Sparsely activated MoE layers (Lepikhin et al., 2020; Fe-

dus et al., 2022; Du et al., 2022; Jiang et al., 2024) further scale FFNs by routing each token to a small subset of expert networks, increasing capacity with limited extra computation. Regarding attention mechanisms, Rotary Positional Embeddings (RoPE) (Su et al., 2024) have largely replaced absolute embeddings to enhance length extrapolation, while Grouped-Query Attention (GQA) (Ainslie et al., 2023) is widely adopted to optimize memory bandwidth during inference. Furthermore, to address the quadratic complexity of standard self-attention, sparse attention mechanisms (Child et al., 2019; Beltagy et al., 2020; Kitaev et al., 2020; Yuan et al., 2025) reduce computation by pruning the attention graph, while linear attention paradigms (Katharopoulos et al., 2020; Choromanski et al., 2020; Gu & Dao, 2023; Han et al., 2023; 2024; Team et al., 2025; Chen et al., 2025) achieve linear scaling with respect to sequence length.

**Depth Pathologies in Pre-Norm Transformers**  While Pre-Norm has become the de facto standard for its optimization robustness (Brown et al., 2020; Touvron et al., 2023a; Dosovitskiy, 2020; Xiong et al., 2020), recent work suggests that very deep Pre-Norm Transformers can exhibit degraded depth utilization(Gromov et al., 2025). A critical limitation of Pre-Norm identified in recent literature is that the unnormalized residual stream can increase in magnitude with depth, creating a scale mismatch between the main path and normalized inputs to residual branches (Sun et al., 2025; Kim et al., 2025).

**LayerNorm Variants**  Layer Normalization (LN) (Ba et al., 2016) is a key component in stabilizing Transformer optimization. Beyond the standard LN, commonly used variants modify the normalization operator itself, such as RMSNorm (Zhang & Sennrich, 2019), ScaleNorm (Nguyen & Salazar, 2019b) and GatedNorm (Qiu et al., 2026). Other approaches alter the placement or frequency of normalization, such as Sandwich-Norm (Ding et al., 2021; Kim et al., 2025) and QK-Norm (Henry et al., 2020), to better regulate activation statistics. In contrast, normalization-free designs like DyT (Zhu et al., 2025c) attempt to remove LN via learnable saturation functions.

## 7. Conclusion

In this work, we propose **SiameseNorm**, a simple yet effective modification to the Transformer residual architecture that combines the optimization stability of Pre-Norm with the stronger representational capacity of Post-Norm. By decoupling LN pathways with negligible overhead, SiameseNorm improves performance while maintaining robust optimization. We view this approach as a promising foundation for future theoretical and empirical studies on new multi-stream paradigms and residual architecture design.

## Acknowledgments

This work is supported in part by the National Key R&D Program of China under Grant 2024YFB4708200, the National Natural Science Foundation of China under Grants 62276150, U24B20173 and U2541227, and the Scientific Research Innovation Capability Support Project for Young Faculty under Grant ZYGXQNJSKYCXNLZCXM-I20.

We thank Zihan Qiu for outstanding insights and generous support, and Zhenda Xie for valuable guidance. We also thank Yifan Pu, Huaqing Zhang and Zichen Liang for helpful discussions, and Zeyu Liu for constructive suggestions on both the codebase and the writing.

## Impact Statement

This paper presents work whose goal is to advance the field of Machine Learning. There are many potential societal consequences of our work, none which we feel must be specifically highlighted here.

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

# A. Appendix

## A.1. Limitations

Despite these promising results, we acknowledge that a limitation of our work, as is common in architectural research, is that our findings are primarily empirical. We do not provide rigorous theoretical explanations for the observed superiority or strict convergence guaranties, as deriving such proofs under realistic training scenarios remains challenging.

## A.2. Comparison with Existing Multi-path Designs

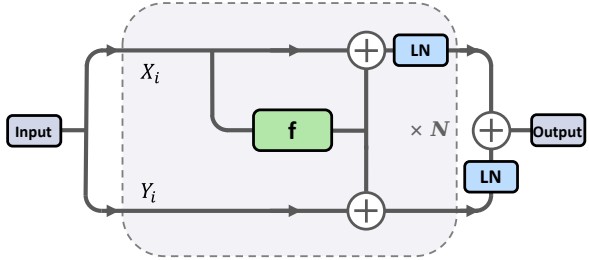

*Figure 7.* Architecture of ResiDual (Xie et al., 2023).

**ResiDual (Xie et al., 2023)**   The work most structurally similar to ours is ResiDual (Xie et al., 2023), as illustrated in Fig. 7. However, a fundamental difference lies in the topology: in ResiDual, the Pre-Norm stream (Y-stream) is not connected to the input of the residual block. This implies that the $Y$-stream acts as a global shortcut that aggregates the output of each residual block directly toward the final output, rather than an active participant in the iterative transformation process. Following the derivation in Equation (10), ResiDual's Jacobian transition matrix is given by:

$$\frac{\partial S_{j+1}}{\partial S_j} = \begin{bmatrix} \mathbf{J}_{\mathrm{LN}_j^X}(\mathbf{I} + \mathbf{J}_{F_j}) & \mathbf{0} \\ \mathbf{J}_{F_j} & \mathbf{I} \end{bmatrix},$$

where $\mathbf{0}$ denotes an all-zero matrix. This implies that the Pre-Norm stream does not receive gradients related to the subsequent residual transformation $F$. Consequently, while the gradients remain relatively stable, they are uninformative. This structural limitation leads to the phenomenon observed in Figure 4: although the model rarely diverges completely, it frequently suffers from severe loss spikes.

**Hyper-Connections(Zhu et al., 2025a)**   Recently, Hyper-Connections and its variant, mHC (Xie et al., 2025), have garnered significant attention within the research community. Similar to our approach, Hyper-Connections attempts to reconcile the Pre-Norm and Post-Norm paradigms. However, as noted in the mHC study, Hyper-Connections encounters training instability. To mitigate this, mHC adopts a design that more closely aligns with the Pre-Norm paradigm.

*Table 4.* Absolute PPL reduction (vs. Pre-Norm baseline) after coupling Pre-Norm with each variant at different learning rates.

| Variant | LR=$4 \times 10^{-4}$ | LR=$1 \times 10^{-3}$ | LR=$2 \times 10^{-3}$ |
|---|---|---|---|
| Post-Pre | 0.18 | 0.12 | 0.17 |
| Hybrid-Pre | **0.64** | **0.41** | **0.41** |

Our empirical evaluations further demonstrate that SiameseNorm exhibits superior training robustness compared to Hyper-Connections. Furthermore, the Hyper-Connections framework is fundamentally compatible with our proposed method. Ablation studies in mHC indicate that $H_{res}$, which facilitates information mixing between parallel streams, is critical for performance gains. In contrast, SiameseNorm omits this design to maintain architectural simplicity. We anticipate that future research will provide a unified perspective on these multi-path paradigms.

## A.3. Choice of Sub-stream Architectures

Preliminary experiments indicate that the performance of SiameseNorm depends on the efficacy of its individual sub-streams. Table 4 compares coupling Pre-Norm with standard Post-Norm versus coupling it with the more advanced HybridNorm. Although standalone Post-Norm performs poorly in this setting, coupling it with Pre-Norm still yields a stable SiameseNorm variant that outperforms both individual baselines. This suggests that the two-stream coupling itself provides a meaningful benefit. Moreover, replacing the Post-Norm sub-stream with the stronger HybridNorm design yields substantially larger gains, indicating that stronger base schemes can further unlock the potential of the SiameseNorm paradigm.

## A.4. Detailed Experimental Settings

The fixed configurations for the 1.3B dense model and the 15A2B MoE model are summarized in Tables 5 and 6, respectively. Note that the learning rate and the total number of training tokens vary in our different experimental setups.

## A.5. Training Loss and Downstream Accuracy Curves

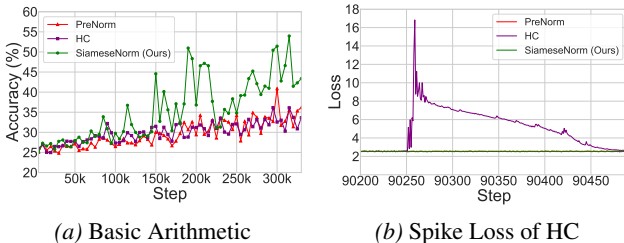

*(a)* Basic Arithmetic     *(b)* Spike Loss of HC

*Figure 8.* Comparison of Pre-Norm (red), HC (purple) and SiameseNorm (green) on pre-training and downstream task using 350B training tokens and 2e-3 learning rate.

*Table 5.* Detailed Experimental Settings for OLMo-1.3B

| Category | Configuration / Value |
| --- | --- |
| **Model architecture** | |
| Number of Layers | 16 |
| Hidden Size | 2048 |
| Attention Heads | 16 |
| Key-Value heads | 16 |
| FFN Intermediate Size | 8192 |
| Activation Function | SwiGLU |
| Tied Word Embeddings | False |
| Position Embedding | RoPE |
| Layer Norm Type | RMSNorm,$\epsilon = 1e - 5$ |
| Vocabulary Size | 50,280 |
| Bias Terms | None |
| QK-Norm | True |
| Initialization | Mitchell (Truncated Normal) |
| **Training setup** | |
| Global Batch Size | 512 |
| Max Sequence Length | 2048 |
| Tokenizer | allenai/OLMo-1B |
| **Optimization** | |
| Optimizer | AdamW ($\beta_1 = 0.9, \beta_2 = 0.95$) |
| Weight Decay | 0.1 |
| LR Scheduler | Cosine with 2,000 warmup steps |
| Final LR Factor | 0.1 |
| Precision | AMP (BF16) |
| Gradient Clipping | 1.0 |

*Table 6.* Detailed Experimental Settings for OLMoE-15A2B

| Category | Configuration / Value |
| --- | --- |
| **Model architecture** | |
| Number of Layers | 24 |
| Hidden Size | 2048 |
| Attention Heads | 16 |
| Key-Value Heads | 16 |
| Number of Experts | 96 |
| Top-$k$ Experts | 8 |
| Expert FFN Ratio | 1 |
| Activation Function | SwiGLU |
| Tied Word Embeddings | False |
| Position Embedding | RoPE |
| Layer Norm Type | RMSNorm |
| Vocabulary Size | 50,280 |
| Embedding Size | 50,304 |
| Bias Terms | None |
| QK-Norm | True |
| Initialization | Normal |
| **Training setup** | |
| Global Batch Size | 512 |
| Max Sequence Length | 2048 |
| Tokenizer | allenai/OLMo-1B |
| **Optimization** | |
| Optimizer | AdamW ($\beta_1 = 0.9, \beta_2 = 0.95$) |
| Learning Rate | $1 \times 10^{-3}$ |
| Weight Decay | 0.1 |
| LR Scheduler | Cosine with 4000 warmup steps |
| Final LR Factor | 0.02 |
| MoE Z-Loss Weight | 0.001 |
| MoE Load-Balancing Loss Weight | 0.01 |

