# OpenReview forum: "SiameseNorm: Breaking the Barrier to Reconciling Pre/Post-Norm"
_ICML.cc/2026/Conference — ICML 2026 regular_

### Official Review · Reviewer_NSCw · 2026-03-11

**Soundness:** 3
**Presentation:** 3
**Significance:** 2
**Originality:** 3
**Overall Recommendation:** 3
**Confidence:** 3

**Summary:**

This paper proposes an improved method for positioning the layernorm layer in the Transformer. Theoretically, it analyzes the weaknesses of commonly used Pre-Norm and Post-Norm methods, arguing that Pre-Norm's gradient propagation is more stable, but it also leads to a decrease in the relative contribution of deeper layers; Post-Norm's magnitude range is converged by layernorm, but it is prone to gradient explosion or vanishing problems. Therefore, the authors combine the characteristics of both methods and propose a dual-path structure, i.e., the SiameseNorm method, which topologically decouples optimization stability from representation stability, thereby achieving stable gradients and representations.

**Compliance With Llm Reviewing Policy:**

Affirmed.

**Final Justification:**

Although some of my concerns regarding the rigor and completeness of the experiments have been partially addressed, the rebuttal does not substantially change my overall assessment, and I therefore maintain my original rating for the current version.

**Key Questions For Authors:**

1. Does SiameseNorm continue to provide benefits when training larger models (e.g., 7B or 13B parameters)? Have the authors explored the scaling behavior of this architecture?

2. Have the authors evaluated SiameseNorm in other Transformer variants, such as Vision Transformers or diffusion-based models?

3. Can the authors provide empirical evidence showing that SiameseNorm indeed improves effective layer utilization compared with standard Pre-Norm Transformers?

**Limitations:**

yes

**Strengths And Weaknesses:**

**Strengths**:

1. Clear theoretical motivation. The paper theoretically analyzes the problems of Pre-Norm and Post-Norm structures in Transformer, and systematically explains the phenomenon that while Transformer training depth increases, the effective depth does not.

2. Simple and effective structural design. By analyzing the two Norm methods, the paper proposes separating representation and gradient propagation into two paths and designs the SiameseNorm structure without increasing the number of model parameters.

3. Sufficient experiments. The paper presents sufficient experiments comparing various methods and parameter designs. Furthermore, the method can still train stably under an aggressive learning rate, demonstrating high practical value in engineering.

4. Clear writing style. The paper is well-structured, clearly presenting the motivation, method, theory, and experiments across various settings.

**Weaknesses**:

1. All experiments are conducted on a 1.3B-parameter language model. While this scale is non-trivial, it is still unclear whether the proposed method maintains its advantages for significantly larger models (e.g., 7B, 13B, or larger). Additional experiments at larger scales would help verify whether the proposed approach provides consistent improvements under realistic large-scale training scenarios.

2. The experiments are restricted to language modeling tasks with a decoder-only Transformer architecture (OLMo). It remains unclear whether SiameseNorm provides similar benefits in other settings, like vision transformer, diffusion transformer, etc. Evaluating the method across a wider range of architectures would strengthen the generality of the proposed approach.

3. One of the key motivations of the paper is that Pre-Norm architectures may suffer from reduced effective depth. However, the paper does not provide direct experimental evidence demonstrating that SiameseNorm improves layer utilization or effective depth. Additional analyses—such as layer contribution measurements, probing experiments, or representation change across layers—would help validate this claim more convincingly.

---

> ### Author Rebuttal · Authors · 2026-03-30
>
> We are grateful to the reviewers for highlighting the clear writing, the conceptual simplicity of our approach, the extensive experimental comparisons and the significant performance gains, alongside the valuable feedback and constructive suggestions provided. Below we address the reviewers' concerns in turn.
>
> ### **A. Experiment Settings**
> Please see our response to Reviewer pg9N for details.
>
> ### **B. Depth Effectiveness**
> To further address the reviewers' feedback, we now provide supplementary experiments of layer contribution measurements conducted on 1.3B models.
>
> 1. **Layer Dropping:** Following established methodologies[1], we retain the final layer, remove the preceding consecutive layers, and examine the loss changes on the C4 validation set.
>
> | Configuration      | Pre-Norm (Loss) | SiameseNorm (Loss) |
> | ------------------ | --------------- | ------------------ |
> | Full Model         | 2.80            | 2.78               |
> | Drop last 1 Layer  | 2.99            | 3.09               |
> | Drop last 2 Layers | 3.23            | 3.60               |
> | Drop last 3 Layers | 3.57            | 4.12               |
> | Drop last 4 Layers | 4.15            | 4.85               |
> | Drop last 5 Layers | 4.98            | 7.04               |
>
> As shown in the table, removing these deep layers leads to a consistently larger loss increase for SiameseNorm than for the Pre-Norm baseline. This suggests that the deeper layers in SiameseNorm make stronger contributions to overall model performance, indicating improved layer utilization compared with the baseline.
>
> 2. **Layer pruning:** We also evaluated the loss changes after removing specific individual layers.
>
> | Layer Pruned | Pre-Norm (Loss) | SiameseNorm (Loss) |
> | ------------ | --------------- | ------------------ |
> | Full Model   | 2.80            | 2.78               |
> | Layer 16     | 3.36            | 3.50               |
> | Layer 15     | 2.99            | 3.09               |
> | Layer 14     | 2.94            | 3.00               |
> | Layer 13     | 2.95            | 3.00               |
> | Layer 12     | 2.95            | 3.00               |
> | Layer 11     | 2.96            | 3.01               |
> | Layer 10     | 2.92            | 2.98               |
> | Layer 9      | 2.89            | 3.69               |
> | Layer 8      | 2.92            | 2.93               |
> | Layer 7      | 2.88            | 2.99               |
> | Layer 6      | 2.90            | 7.03               |
> | Layer 5      | 2.90            | 3.72               |
> | Layer 4      | 3.01            | 3.34               |
> | Layer 3      | 2.98            | 3.42               |
> | Layer 2      | 3.96            | 10.42              |
> | Layer 1      | 4.61            | 10.39              |
>
> As the results show, regardless of which layer is pruned, SiameseNorm consistently suffers a larger performance drop than the Pre-Norm baseline. This suggests that individual layers in SiameseNorm contribute more to overall model performance, indicating stronger layer utilization.
>
> ### **C. Ablation Studies**
> Please see our response to Reviewer xhSU for details. Through additional experiments, we explicitly demonstrate that this design successfully inherits the distinct advantages of both Pre-Norm and Post-Norm.
>
> ### **D. Computational Overhead**
> Please see our response to Reviewer Lz9y for details. Our method introduces minimal computational overhead, both theoretically and empirically.
>
> ---
>
> ### Response to Weakness 1, Weakness 2, Question 1, and Question 2
> **A. Experiment Settings** address these concerns. We have added extensive experiments to further support our claims, including evaluations across varying depths, scaling up to a **15B**-parameter large model, and validating our method on both **ViT** and **DiT** architectures. Across all these settings, SiameseNorm consistently outperforms the Pre-Norm baseline. Notably, as a plug-and-play replacement, it achieves significant improvements on both ViT and DiT, yielding a +1.5% gain in Top-1 image-classification accuracy and a 3.87 reduction in FID, respectively.
>
> ### Response to Weakness 3 and Question 3
>  **B. Depth Effectiveness** addresses these concerns. By introducing layer contribution measurements, we demonstrate that our method achieves a higher layer utilization rate compared to Pre-Norm.
>
> We thank the reviewer again for the thoughtful comments, and we hope our responses have addressed the reviewer’s concerns and clarified the questions raised.
>
> [1] The Unreasonable Ineffectiveness of the Deeper Layers.

---

> > ### Author Rebuttal · Reviewer_NSCw · 2026-04-03
> >
> > Thanks to the authors for the detailed rebuttal, which has addressed most of my concerns. However, as there are still some remaining issues, including limited evidence on scalability to larger models and insufficiently direct validation of the claimed improvements in layer utilization. Overall, I will maintain my current rating.

---

> > > ### Author Response · Authors · 2026-04-03
> > >
> > > We appreciate the reviewer acknowledging that our rebuttal has addressed most of the prior concerns. Regarding the remaining issues on scalability and layer utilization, we would like to further clarify that the current evidence already meets the standard commonly adopted in large-scale Transformer research, and provides consistent support for our claims.
> > >
> > > ### 1. Significance of the Contribution
> > >
> > > Improving LayerNorm design beyond Pre-Norm has long been a fundamental challenge in Transformer research. Owing to its strong optimization stability and competitive performance, Pre-Norm has become the **dominant** design choice in modern Transformers, even though it may sacrifice some representational benefits associated with Post-Norm. In particular, Post-Norm introduces **valuable nonlinearity** at the residual summation, which can enhance representation dynamics, but existing Post-Norm-style designs are often difficult to train stably. Our goal is therefore to develop a **simple, plug-and-play** improvement over Pre-Norm that achieves better performance without sacrificing its stability. We believe this makes our contribution meaningful rather than incremental: SiameseNorm addresses a long-standing architectural tension through a minimal modification, introduces little overhead, and delivers consistent gains across scales, depths, and modalities.
> > >
> > > ### 2. Scalability and Generality
> > >
> > > We would like to clarify that our empirical evidence already covers a broad range of scales and architectures. Our experiments span from **5M parameters (DeiT-Tiny)** to **15B parameters (MoE)**, with a total compute cost across all experiments equivalent to over 60,000 A100 GPU hours. This level of empirical coverage is already broad and comparable to that of several highly influential prior works [1][2][3][4].
> > >
> > > We also evaluated SiameseNorm beyond decoder-only language models. As a direct plug-and-play replacement, it improves Top-1 accuracy by +1.5% on DeiT and reduces FID by 3.87 on DiT. We believe such cross-architecture and cross-modality gains provide meaningful evidence that the benefit comes from a general architectural improvement rather than from task-specific tuning.
> > >
> > > ### 3. Effective Depth and Layer Utilization
> > >
> > > We agree that effective depth should be supported by direct empirical evidence. To address this, we added **layer dropping** and **layer pruning** experiments. We would like to emphasize that such removal-based analyses are standard ways[5][6] to measure layer contribution and utilization in deep models. Under these evaluations, SiameseNorm consistently suffers **larger degradation when layers are removed**, indicating that its layers make stronger contributions to the final model behavior.
> > >
> > > In addition, our paper already provides further evidence consistent with improved utilization. We find that the **Post-Norm stream has a larger influence on the final output**, suggesting that the branch associated with stronger representation dynamics is being effectively used rather than bypassed. Moreover, SiameseNorm shows especially strong gains on the **arithmetic task**, which places greater demand on sequential reasoning.
> > >
> > > Taken together, these results support our claim from multiple angles: deeper-layer removal causes larger degradation, the Post-Norm stream contributes more strongly to the output, and tasks requiring stronger sequential reasoning benefit more substantially. All of these are consistent with improved effective depth relative to the Pre-Norm baseline.
> > >
> > > We fully agree that even broader scaling studies and additional probing analyses would further strengthen the paper, and we will continue pursuing them. At the same time, since the reviewer noted that our rebuttal has addressed most concerns, we hope the reviewer may consider taking this into account in the final score.
> > >
> > > Thank you again for the careful reading and helpful comments.
> > >
> > > [1] Hyper-Connections.
> > >
> > > [2] Gated Attention for Large Language Models: Non-linearity, Sparsity, and Attention-Sink-Free.
> > >
> > > [3] mHC: Manifold-Constrained Hyper-Connections.
> > >
> > > [4] DeepNet: Scaling Transformers to 1,000 Layers.
> > >
> > > [5] The Unreasonable Ineffectiveness of the Deeper Layers.
> > >
> > > [6] Are Sixteen Heads Really Better than One?

---

### Official Review · Reviewer_xhSU · 2026-03-12

**Soundness:** 2
**Presentation:** 3
**Significance:** 2
**Originality:** 2
**Overall Recommendation:** 4
**Confidence:** 3

**Summary:**

The paper concerns Transformers normalization design, focusing on the trade-off between Pre-Norm and Post-Norm. Authors propose SiameseNorm, a dual-stream residual architecture in which a Pre-Norm-like stream preserves a clean identity gradient path while a Post-Norm-like stream maintains bounded representation scales.

**Compliance With Llm Reviewing Policy:**

Affirmed.

**Final Justification:**

Although this work appears to need more rigorous proofs and evidence, I find the authors' last response reasonable, and thus, would like to remain positive.

**Key Questions For Authors:**

- The final method combines several mechanisms (normalized fusion, depth-wise scaling). It would be useful to clarify whether the observed stability enhancements stem primarily from the Siamese topology itself or from these auxiliary stabilization techniques. In particular, can the authors report the results of Siamese topology without both normalized inputs and depth-wise scaling?
- The analysis of optimization stability mainly relies on the total gradient norm. Given that the paper's central claim concerns gradient propagation across depth, it would be helpful to report layer-wise gradient norm statistics. Could the authors provide it or clarify why such analysis was not included?

**Limitations:**

yes

**Strengths And Weaknesses:**

Strengths
- The proposed two-stream residual architecture is conceptually simple and can be integrated into existing Transformer models with minimal modification.
- The experiments show that SiameseNorm maintains training stability under aggressive learning rates where several Post-Norm variants diverge, while also improving perplexity and downstream task scores
- The paper includes ablations about Siamese topology, depth-wise scaling, and normalized input. These experiments help clarify which components contribute to stability and performance.

Weakness
- While the proposed dual-stream architecture is intuitively appealing, the paper does not clearly explain why this design fundamentally resolves the Pre-Norm/Post-Norm trade-off.
- Although the authors claim negligible overhead, the paper does not provide detailed analysis of activation memory cost, inference latency, or training FLOPs.

---

> ### Author Rebuttal · Authors · 2026-03-30
>
> We are grateful to the reviewers for highlighting the clear writing, the conceptual simplicity of our approach, the extensive experimental comparisons and the significant performance gains, alongside the valuable feedback and constructive suggestions provided. Below we address the reviewers' concerns in turn.
>
> ### **A. Experiment Settings**
>
> Please see our response to Reviewer pg9N for details.
>
> ### **B. Depth Effectiveness**
>
> Please see our response to Reviewer NSCw for details.
>
> ### **C. Ablation Studies**
>
> We agree that this part should be presented more clearly. Below, we clarify it from three aspects.
>
> 1. The Siamese design is the **central contribution** of our method. By allowing the residual transform $F_i$ to receive two complementary types of gradients, our method provides a structural mechanism for better balancing the respective advantages of Pre-Norm and Post-Norm, **a claim that is further supported by our experiments:**
>
> - First, the results already included in the paper show that the Siamese topology itself is effective. In Table 3, row2 vs. row6 shows that even after ablating the strong auxiliary components, the Siamese design still reduces perplexity by **0.22**, corresponding to about **0.02** reduction in loss. In LLM pretraining, this scale is already considered significant [1][2][3].
> - Second, the new rebuttal results further support that it inherits strengths from both paradigms. In **B. Depth Effectiveness**, SiameseNorm shows higher layer utilization than Pre-Norm, reflecting a property commonly associated with **Post-Norm**. In addition, under a standard 2K warm-up, SiameseNorm **without** depth-scaling and HybridNorm with depth-scaling achieve similar perplexities (10.68 vs. 10.65). However, when the warm-up is shortened, HybridNorm diverges with **300** warm-up steps, whereas SiameseNorm remains stable even with **no warm-up**, suggesting robustness closer to **Pre-Norm** [4]. Together, these results support that the Siamese design helps reconcile the Pre-Norm/Post-Norm trade-off in practice.
>
> 2. Depth-scaling and normed-input are introduced mainly to make SiameseNorm **plug-and-play** under the standard Pre-Norm recipe. Their role is to make the Siamese design easier to adopt without hyperparameter retuning. **This is also supported by experiments:**
>
> - The ablation results in Table 3 of the paper demonstrate the effectiveness of these two components in SiameseNorm.
> - The additional **ViT** and **DiT** results in **A. Experiment Settings** further show that the full design brings consistent gains under unchanged hyperparameters. Specifically, SiameseNorm achieves a +1.5% gain in Top-1 image-classification accuracy and a 3.87 reduction in FID, supporting its plug-and-play property.
> - Without both normed-input and depth-scaling, the Siamese topology remains **stable** and reaches a PPL of 10.88 (0.2 worse than the version with normed-input).
>
> 3. We clarify that the specific choice of Pre-Norm and Post-Norm variants is not the main focus of this paper, as competitive baselines and extensive ablations already support the validity of these three components.
>
> ### **D. Computational Overhead**
>
> Please see our response to Reviewer Lz9y for details. Our method introduces minimal computational overhead, both theoretically and empirically.
>
> ---
>
> ### Response to Weakness 1 and Question 1
>
> **C. Ablation Studies** address this concern.
>
> ### Response to Weakness 2
>
> **D. Computational Overhead** addresses this concern.
>
> ### Response to Question 2
>
> - In practice, training instability is often accompanied by gradient spikes or outliers, which are directly reflected in the total gradient norm. We therefore use it as our primary indicator of optimization stability.
> - By contrast, layer-wise gradient norms can vary substantially across layers and components (e.g., Attention, FFN, and Layer-Norm parameters), making their raw absolute values hard to relate directly to stability or performance. For this reason, we did not include them in the current paper.
>
>
> We thank the reviewer again for the thoughtful comments, and we hope our responses have addressed the reviewer’s concerns and clarified the questions raised.
>
> [1] Gated Attention for Large Language Models: Non-linearity, Sparsity, and Attention-Sink-Free.
>
> [2] mHC: Manifold-Constrained Hyper-Connections.
>
> [3] Attention Residuals.
>
> [4] On Layer Normalization in the Transformer Architecture.

---

> > ### Author Rebuttal · Reviewer_xhSU · 2026-04-03
> >
> > Thank you for the rebuttal. I found the responses helpful overall, and the additional ablations and clarifications were appreciated.
> >
> > A few points still seem worth clarifying. While the work presents an intuitive design and thorough ablation studies, I still feel that the conceptual explanation of how SiameseNorm inherits only the advantages of both Pre-Norm and Post-Norm is not yet fully clear. In particular, it remains somewhat unclear whether the proposed design truly combines the benefits of the two paradigms, or whether it instead lands at an intermediate compromise between them. A clearer articulation of this point would further strengthen the contribution of SiameseNorm.
> >
> > That said, I still value this paper positively overall. I appreciate that it achieves meaningful performance gains without introducing noticeable additional cost, and that the design has been examined through extensive ablations. For these reasons, I am maintaining my overall positive assessment.
> >
> > I also still have a few follow-up questions regarding the use of total gradient norm as a stability indicator.
> >
> > (1) In the rebuttal, the authors explain that they relied on total gradient norm because different layers can have very different scales. However, it still seems feasible to compare the frequency or variance of gradient spikes within the same layer across methods. In particular, examining layers closer to the input might provide more direct evidence regarding stability.
> >
> > (2) Relatedly, if the scale differs substantially across layers, total gradient norm may fail to adequately reflect instability arising in smaller-scale layers.
> >
> > For this reason, I would appreciate a bit more clarification on why the total gradient norm should be considered an appropriate indicator of stability in this setting.

---

> > > ### Author Response · Authors · 2026-04-03
> > >
> > > We sincerely thank the reviewer for the positive overall assessment and for recognizing the meaningful performance gains, negligible additional cost, and extensive ablations. We also appreciate these follow-up questions, which help us clarify two important points.
> > >
> > > (1) On whether SiameseNorm truly combines the benefits of Pre-Norm and Post-Norm, or instead represents an intermediate compromise
> > >
> > > Our clarification is as follows.
> > > - **Broader functional potential**. SiameseNorm has a broader parameter space than either Pre-Norm or Post-Norm alone. As discussed in the paper, the Siamese design can recover Pre-Norm-like or Post-Norm-like behavior as special cases, giving it a potentially higher performance ceiling than either paradigm individually.
> > > - **Richer optimization signals**.  Our design introduces complementary gradient information that is unavailable in either paradigm alone. Specifically, the two-stream structure allows the shared residual transform to receive gradient signals from both an identity-preserving path and a scale-bounded path. Intuitively, this is analogous in spirit to how residual connections in ResNet [1] expose gradient propagation paths of different effective depths, thereby improving optimization. Similarly, rather than forcing a single-stream architecture to trade off between these two desirable properties, our design creates a structure in which both types of optimization signals can be preserved and exploited jointly.
> > >
> > > That said, we fully acknowledge that the optimization behavior of modern Transformers is **highly complex**, and the commonly accepted strengths and weaknesses of Pre-Norm and Post-Norm are themselves largely empirical. Under this premise, it is difficult to rigorously prove whether our design fully inherits the benefits of both paradigms or instead lies at some intermediate point. Our claim is therefore mainly empirical: through extensive ablations, we show that SiameseNorm achieves clear performance advantages together with very high stability.
> > >
> > > (2) On why total gradient norm is an appropriate stability indicator
> > >
> > > In our paper, we use total gradient norm together with the loss curve as practical indicators of training stability, following both empirical experience and prior work [2][3][4]. In particular, [2] reports that even small loss bumps are consistently accompanied by significant gradient spikes, sometimes reaching magnitudes up to $1000\times $larger than typical gradients.
> > >
> > > Our main intuition is the following: The total gradient norm reflects the overall update intensity of the model at a given step. A sudden spike in total gradient norm often means an abnormally large parameter update, which can directly push training out of a stable region and lead to a loss spike. For this reason, we use total gradient norm and the loss curve as practical indicators of stability.
> > >
> > > To further address the reviewer’s concern, we additionally provide visualizations of layer-wise gradient norms for the projection layers in the Attention and FFN modules:
> > > https://anonymous.4open.science/r/layerwise-gradient-norms
> > >
> > > These plots show that:
> > > - Loss spikes coincide with spikes in both total gradient norm and layer-wise gradient norms;
> > > - SiameseNorm generally has smaller gradient norms than Pre-Norm, while both methods maintain stable loss curves throughout training.
> > > In conclusion, our point is not that total gradient norm is the only possible indicator, but rather that it is a practical and informative aggregate signal, and in our setting it is consistent with both the loss behavior and the additional layer-wise evidence.
> > >
> > > We thank the reviewer again for these constructive questions and hope these clarifications make our position clearer.
> > >
> > > [1] Identity Mappings in Deep Residual Networks.
> > >
> > > [2] SPAM: Spike-Aware Adam with Momentum Reset for Stable LLM Training.
> > >
> > > [3] Spike No More: Stabilizing the Pre-training of Large Language Models.
> > >
> > > [4] On Layer Normalization in the Transformer Architecture.

---

### Official Review · Reviewer_Lz9y · 2026-03-12

**Soundness:** 3
**Presentation:** 2
**Significance:** 2
**Originality:** 2
**Overall Recommendation:** 2
**Confidence:** 3

**Summary:**

It provides a Jacobian motivation for why the two-stream transition matrix inherits Pre-Norm-like and Post-Norm-like gradient components in parallel, then evaluates a practical instantiation that couples Pre-Norm with HybridNorm blocks. On 1.3B OLMo-style language models trained on FineWeb-Edu, SiameseNorm is reported to remain stable at learning rates where several Post-Norm baselines diverge, while improving perplexity and average downstream score, with gains on the Arithmetic task. The main contribution is an interesting architectural proposal plus empirical evidence that a shared-parameter dual-stream topology may improve the stability/performance frontier.

**Compliance With Llm Reviewing Policy:**

Affirmed.

**Final Justification:**

Because the author’s responses to questions about the theoretical aspects and working principles in this paper were too vague, I am keeping the current score.

**Key Questions For Authors:**

Please refer to the above weaknesses.

**Limitations:**

NO. Please refer to the above weaknesses.

**Strengths And Weaknesses:**

Strengths:

1. Table 1 is a good asset. Across the reported 100B- and 350B-token runs, SiameseNorm is competitive or best on perplexity and average downstream score, and its Arithmetic gains at the aggressive learning rate are substantial. Relative to normalization/topology baselines, the reported margins are meaningful enough to justify attention from the community [2][4][5][6].
2. The two-stream construction is easy to understand and well aligned with the motivating tension between identity-gradient preservation and bounded representation scale. Sharing the residual block parameters across streams is an elegant design choice, because it avoids turning the method into “bigger model wins” comparison and keeps the proposal closer to a topology contribution than a capacity contribution [1][2][3].

Weaknesses:

1. Sec.2 and Sec.3 are NOT a rigorous theory. The paper makes sweeping statements such as “mathematically impossible” or “strictly generalizes” the prior paradigms, yet the derivations do not establish formal impossibility, stability bounds, convergence guarantees, or equivalence conditions for the claimed special cases. In particular, inspecting the block Jacobian structure is not enough to conclude that the full coupled system inherits the best properties of both paradigms in a rigorous sense [1][5][6].
2. This is the most serious empirical issue for me. Appendix Table 4 indicates that the main model has only **16 layers**, which is difficult to reconcile with its repeated framing around solving depth pathologies, restoring effective depth, or overcoming the limitations of very deep Pre-Norm Transformers. The arithmetic improvements and hidden-state-norm plots are suggestive, but they are not a direct substitute for experiments on actually deeper stacks (or for direct depth-utilization analyses such as layer pruning or per-layer contribution studies) [5][9].
3. Its good performance is not just the Siamese topology. It also adds a normalized fused input, depth-wise scaling, and a HybridNorm-specific stream design with additional learnable mixing. Table 3 already shows that some of these choices materially rescue the single-stream baseline, so the causal claim that the final gain primarily comes from the Siamese decomposition is not yet convincingly isolated. This matters because extra normalization and residual-scaling tricks are already known to be powerful [2][5][7].
4. The claim of “negligible computational overhead” is unclear. Even if parameter overhead is small, maintaining two hidden streams has activation-memory and possibly throughput implications that are not quantified. And the paper does not report variance across seeds, confidence intervals, or a clearer account of whether the hyperparameter search budget was comparable across methods [4][7][8].

[1] On Layer Normalization in the Transformer Architecture
[2] HybridNorm: Towards Stable and Efficient Transformer Training via Hybrid Normalization
[3] Mix-LN: Unleashing the Power of Deeper Layers by Combining Pre-LN and Post-LN
[4] Hyper-Connections
[5] DeepNet: Scaling Transformers to 1,000 Layers
[6] ResiDual: Transformer with Dual Residual Connections
[7] NormFormer: Improved Transformer Pretraining with Extra Normalization
[8] ReZero is All You Need: Fast Convergence at Large Depth
[9] The Unreasonable Ineffectiveness of the Deeper Layers

---

> ### Author Rebuttal · Authors · 2026-03-30
>
> We are grateful to the reviewers for highlighting the clear writing, the conceptual simplicity of our approach, the extensive experimental comparisons and the significant performance gains, alongside the valuable feedback and constructive suggestions provided. Below we address the reviewers' concerns in turn.
>
> ### **A. Experiment Settings**
> Please see our response to Reviewer pg9N for details.
>
> ### **B. Depth Effectiveness**
> Please see our response to Reviewer NSCw for details.
>
> ### **C. Ablation Studies**
> Please see our response to Reviewer xhSU for details.
>
> ### **D. Computational Overhead**
> Our method introduces minimal computational burden and the specific quantitative results are as follows:
>
> 1. Extremely low Theoretical overhead: Compared to the baseline model, the increases in both parameter count and FLOPs during training and inference are strictly controlled within 0.1%.
>
> 2. High efficiency in large models: In the 15B parameter MoE model in **A. Experiment Settings**, our method decreases overall training speed by only 0.5%. Furthermore, it introduces merely a 2% activation memory overhead, which is negligible compared to the Attention and MoE components.
>
> ---
>
> ### Response to Weakness 1
> We thank the reviewer for this important comment. We agree that the presentation in Sec. 2 and Sec. 3 was not sufficiently precise, and that some of our original wording overstated the level of rigor of the analysis. We will revise these sections accordingly. We clarify two points below:
>
> - **Structural intuition rather than formal proof.** Sec. 2 and Sec. 3 are not intended to establish impossibility results, stability bounds, or convergence guarantees. Instead, we formalize our motivation through a Jacobian-based perspective, together with empirical observations from prior work, to provide intuition for the different optimization behaviors of Pre-Norm and Post-Norm, and to motivate why a two-stream design that couples Pre-Norm and Post-Norm may help reconcile their respective advantages.
> - **Empirical evidence rather than theoretical guarantees.** Importantly, the effectiveness of our method is established empirically rather than through rigorous proof of convergence or optimality. Across both our main large-scale pretraining setting and the supplementary results, SiameseNorm consistently demonstrates strong stability and improved performance. In particular, as shown in **C. Ablation Studies**, through additional experiments we provide further empirical evidence that our design exhibits key advantages commonly associated with both Pre-Norm and Post-Norm. We will revise the paper accordingly to make this scope explicit.
>
> ### Response to Weakness 2
> **A. Experiment Settings** and **B. Depth Effectiveness** address this concern:
>
> - We clarify the motivation behind our original experimental setup.
> - We add experiments with varying model depths and layer-utilization analyses, and the results are consistent with our previous findings.
>
> ### Response to Weakness 3
> **C. Ablation Studies** addresses this concern:
>
> - Notably, even after ablating these already well-established and powerful components, the smallest improvement brought by our Siamese design in Table 3 is still a 0.22 PPL gain, corresponding to a **0.02** reduction in loss. In LLM pretraining, this scale is already considered significant[1][2][3].
> - Through additional experiments, we provide further evidence that our design inherits key advantages of both Pre-Norm and Post-Norm, in contrast to prior approaches that stabilize Post-Norm within a single-stream architecture.
>
> ### Response to Weakness 4
> **D. Computational Overhead** addresses this concern. Additionally:
>
> - Variance across seeds and confidence intervals: Repeated full-scale pretraining runs at this scale are extremely costly. Accordingly, as is common in large-scale pretraining studies[1][2][3], we report results under a fixed seed rather than across multiple seeds. Following OLMo, we use the default seed 6198 throughout for fair comparison.
> - Hyperparameter search budget: We do not perform extra hyperparameter search. Instead, we fully follow the standard Pre-Norm OLMo setup and compare methods under different learning rates. Our goal is to evaluate whether SiameseNorm is a **plug-and-play** improvement under mainstream pretraining settings rather than one that depends on carefully tailored tuning. The results across learning rates and additional experiments consistently support this point.
>
> We thank the reviewer again for the thoughtful comments, and we hope our responses have addressed the reviewer’s concerns and clarified the questions raised.
>
> [1] Gated Attention for Large Language Models: Non-linearity, Sparsity, and Attention-Sink-Free.
>
> [2] mHC: Manifold-Constrained Hyper-Connections.
>
> [3] Attention Residuals.

---

> > ### Author Rebuttal · Reviewer_Lz9y · 2026-04-04
> >
> > Thank you to the authors for the rebuttal; it partially addresses my concerns. However, your response to W1 is too general, and having a more substantial theoretical explanation and clarification would further enhance the contribution and depth of this paper. Furthermore, how SiameseNorm has only the advantages of both Pre-Norm and Post-Norm is not yet fully clear. I prefer to maintain the score.

---

> > > ### Author Response · Authors · 2026-04-04
> > >
> > > We deeply appreciate the reviewer’s continued engagement and constructive push for a more substantial theoretical clarification. To further clarify why SiameseNorm can better reconcile the desirable properties of Pre-Norm and Post-Norm, we provide a more concrete mechanistic perspective together with our structural design principles.
> > >
> > > **Broader functional potential.** SiameseNorm has a broader parameter space than either Pre-Norm or Post-Norm alone. As discussed in the paper, the Siamese design can recover Pre-Norm-like or Post-Norm-like behavior as special cases, giving it a potentially higher functional ceiling than either paradigm individually.
> > >
> > > **Richer optimization signals.** More importantly, our design introduces complementary gradient information that is unavailable in either paradigm alone. Specifically, the two-stream structure allows the shared residual transform to receive gradient signals from both an identity-preserving path and a scale-bounded path. Intuitively, this is analogous in spirit to how residual connections in ResNet [1] expose gradient propagation paths of different effective depths, thereby improving optimization. Similarly, rather than forcing a single-stream architecture to trade off between these two desirable properties, our design creates a structure in which both types of optimization signals can be preserved and jointly exploited.
> > >
> > > That said, we fully agree that this does not constitute a formal proof that SiameseNorm strictly inherits all advantages of both paradigms. More precisely, our claim is that SiameseNorm provides a structural mechanism that makes such a reconciliation possible, while the evidence for its effectiveness is primarily empirical. Since the optimization dynamics of modern Transformers are **highly complex**, and even the commonly accepted strengths and weaknesses of Pre-Norm and Post-Norm are themselves **largely established empirically**, it is difficult to formulate a fully rigorous proof at present. Under this positioning, our main claim is empirical: through extensive ablations and supplementary experiments, we show that SiameseNorm consistently achieves strong stability together with clear performance gains.
> > >
> > > **Significance of the contribution.** Improving LayerNorm design beyond Pre-Norm has long been a fundamental challenge in Transformer research. Owing to its strong optimization stability and competitive performance, Pre-Norm has become the dominant design choice in modern Transformers, even though it may sacrifice some representational benefits associated with Post-Norm. In particular, Post-Norm introduces valuable nonlinearity at the residual summation, which can enrich representation dynamics, but existing Post-Norm-style designs are often difficult to train stably. Our goal is therefore to develop a simple, plug-and-play improvement over Pre-Norm that achieves better performance without sacrificing its stability. We believe this makes our contribution meaningful rather than incremental: SiameseNorm addresses a long-standing architectural tension through a minimal structural modification, introduces negligible overhead, and delivers consistent gains across scales, depths, and modalities.
> > >
> > > We thank the reviewer again for the thoughtful comments and constructive engagement. We hope that the above clarifications, together with the additional empirical evidence provided in the rebuttal, help address the remaining concerns on both the theoretical positioning and the significance of the contribution. In light of these clarifications and results, we respectfully hope the reviewer will reconsider the current assessment.
> > >
> > > [1] Identity Mappings in Deep Residual Networks.

---

### Official Review · Reviewer_pg9N · 2026-03-18

**Soundness:** 3
**Presentation:** 4
**Significance:** 2
**Originality:** 2
**Overall Recommendation:** 4
**Confidence:** 4

**Summary:**

This paper proposes SiameseNorm, a two-stream Transformer normalization design that combines a Pre-Norm and a Post-Norm-like stream with shared residual-block parameters. The goal is to retain the optimization stability of Pre-Norm while also benefiting from the bounded-representation behavior that of Post-Norm. With theoretical motivation, controlled pre-training experiments on 1.3B-scale models across multiple learning rates, and ablations showing that the proposed method and its design choices improve stability and perplexity relative to several baselines.

**Compliance With Llm Reviewing Policy:**

Affirmed.

**Key Questions For Authors:**

Q1) Could the authors clarify what should be regarded as the final recipe and which components should be viewed as essential?

Q2) How stable is the paper’s main empirical conclusion under a broader optimization sweep than the currently reported settings?

Q3) To what extent should the current findings be interpreted as specific to the 1.3B OLMo-style setting, rather than as indicative of a broader pattern?

**Limitations:**

yes

**Strengths And Weaknesses:**

### Strengths
The paper is clearly written and easy to follow. The motivation is standard for this line of work, but the proposed two-stream design is presented in a reasonably clear way, and the empirical section makes the main comparisons easy to track.

### Weaknesses
The empirical support is somewhat narrow relative to the scope of the claims, as the evaluation is largely centered on a single 1.3B OLMo-style setting. The ablation study is helpful, but it still leaves some ambiguity about how much of the final gain should be attributed to SiameseNorm itself versus the additional components included in the final recipe. In addition, given the paper’s positioning around the Pre-/Post-Norm trade-off, a direct comparison with closely related baselines such as sandwich-norm (Peri-LN) and SpanNorm would have strengthened the empirical case.

---

> ### Author Rebuttal · Authors · 2026-03-30
>
> We are grateful to the reviewers for highlighting the clear writing, the conceptual simplicity of our approach, the extensive experimental comparisons and the significant performance gains, alongside the valuable feedback and constructive suggestions provided. Below we address the reviewers' concerns in turn.
>
> ###  **A. Experiment Settings**
> Improving LayerNorm design beyond Pre-Norm has long been a fundamental challenge. Meanwhile, owing to its strong optimization stability and competitive performance, Pre-Norm has become the **dominant** design choice in modern Transformers. Thus, our goal is to develop a **plug-and-play improvement** over Pre-Norm that achieves better performance without sacrificing its stability. The submitted version already provides strong initial evidence under **standard** Pre-Norm settings. Following the reviewers’ suggestions, we further add experiments in four directions:
>
> 1. **Varying depths.** Under a fixed training budget of 12B tokens and a nearly constant model size of 390M, we compared Pre-Norm and SiameseNorm at different depths with LR=1e-3:
> |Layers|Hidden Dimension|Pre-Norm PPL|Siamese-Norm PPL|PPL Reduction|
> |---|---|---|---|---|
> |17|1024|17.23|15.69|1.54|
> |33|768|17.29|**15.64**|1.65|
> |80|512|18.02|15.98|**2.04**|
>
> SiameseNorm consistently outperforms the Pre-Norm baseline across all depths. This advantage is particularly pronounced in deep and narrow architectures, achieving a maximum ppl reduction of 2.04 at 80 layers, which provides further evidence of its effectiveness in deeper networks.
>
> 2. **Larger-scale**. We conducted Mixture-of-Experts (MoE) experiments on a 24-layer model with 15B total / 2B active parameters, trained on 100B tokens with LR=1e-3:
> |Name|PPL|ARC-E|ARC-C|HS|OBQA|PIQA|WG|Arith.|Avg Score|
> |---|---|---|---|---|---|---|---|---|---|
> |Pre-Norm|7.92|78.9|47.8|69.4|45.2|77.1|61.8|48.6|61.26|
> |SiameseNorm|**7.76**|78.9|**48.8**|**69.9**|44.0|**77.9**|**63.8**|**58.2**|**63.07**|
>
> SiameseNorm consistently improves perplexity and overall downstream performance under this setting.
>
> 3. **Additional baselines and ablation experiments.**  We provide additional results on baselines and ablation experiments:
>
>   - Peri-LN[1]: PPL 10.89 at LR 2e-3, matching Pre-Norm, with a slightly higher average downstream score (53.24 vs. 52.94).
>   - SpanNorm[2]: diverges at LR 2e-3, and reaches PPL 11.00 / 10.86 at LR 4e-4 / 1e-3.
>   - Ablation experiments: Please see **C. Ablation Studies**.
>
> Consistent with our original conclusions, these supplementary experiments show that Post-Norm variants are highly hyperparameter-sensitive, while our method achieves clear advantages over competing methods.
>
> 4. **Other Transformer variants**. We also include experiments on other Transformer architectures:
> - Vision Transformers: On DeiT [3], by directly combining Pre-Norm and Post-Norm into our SiameseNorm design, the Top-1 accuracy improves from 72.2% to 73.6% on DeiT-Tiny and from 79.8% to 81.3% on DeiT-Small.
> - Diffusion Transformer[4]: Similarly, on DiT-L/4 (24 layers), using the same Pre-Norm + Post-Norm Siamese design, SiameseNorm achieves a lower FID of 41.34 than the Pre-Norm baseline of 45.21.
>
> In summary, the observations from the additional experiments are largely consistent with our original findings, which further strengthens support for our method. We will include all of these results in the revised version.
>
> ### **B. Depth Effectiveness**
> Please see our response to Reviewer NSCw for details.
>
> ### **C. Ablation Studies**
> Please see our response to Reviewer xhSU for details. Through additional experiments, we provide additional evidence that this design inherits key advantages of both Pre-Norm and Post-Norm.
>
> ### **D. Computational Overhead**
> Please see our response to Reviewer Lz9y for details. Our method introduces minimal computational overhead, both theoretically and empirically.
>
> ---
>
> ### Response to Q1
>  **C. Ablation Studies** addresses this concern:
>
> 1. Our core contribution is the Siamese design, whose effectiveness is further supported by additional ablations.
> 2. Depth-scaling and normed-input are mainly introduced to make the architecture plug-and-play under standard Pre-Norm recipes.
> 3. The specific choice of Pre-Norm and Post-Norm variants is not the main focus of this paper.
>
> ### Response to Q2&Q3
>  **A. Experiment Settings** addresses these concerns. SiameseNorm consistently yields significant improvements across different hyperparameter settings, model sizes, depths, and modalities, demonstrating the generality of our method.
>
> We hope our responses have addressed the reviewer’s concerns and clarified the questions raised.
>
> [1] Peri-LN: Revisiting Normalization Layer in the Transformer Architecture.
>
> [2] SpanNorm: Reconciling Training Stability and Performance in Deep Transformers.
>
> [3] Training data-efficient image transformers & distillation through attention.
>
> [4] Scalable Diffusion Models with Transformers.

---

> > ### Author Rebuttal · Reviewer_pg9N · 2026-04-06
> >
> > The rebuttal helps clarify the authors’ intended interpretation of the paper, but my concerns are not fully resolved. In particular, it is still somewhat unclear how readers should interpret the final recipe. While the response suggests that the Siamese design is the core contribution and the other choices are mainly for compatibility, the relative role of these components remains insufficiently separated. I also encourage the authors to calibrate the scope of the paper’s claims carefully in the revision, and to clarify more explicitly which parts of the recipe are essential versus auxiliary.

---

### Decision · Program_Chairs · 2026-04-30

**Decision:**

Accept (regular)

**Comment:**

This paper proposes a variant of normalization that bridges pre-norm and post-norm designs through a novel siamese stream architecture. The initial reviews were mixed: two leaned toward acceptance and two toward rejection, and the overall assessment remained largely unchanged after the rebuttal phase. Accordingly, the AC has carefully reviewed the paper, the reviews, and the rebuttal, and has additionally examined the work independently to ensure a fair and thorough evaluation.

The major concerns raised by the reviewers are largely consistent: (1) the method's scalability and limited evaluation (pg9N, Lz9y, NSCw); (2) unclear rationale behind the observed improvements, whether they stem from normalization itself or additional architectural components (pg9N, Lz9y, xhSU); and (3) added computational overhead (Lz9y, xhSU). Additional weaknesses include the rigor of theoretical justification (Lz9y), the need for deeper analysis (xhSU), and limited validation beyond language modeling (NSCw).

After the rebuttal, the AC finds that several key concerns have been substantially addressed. In particular, scalability and evaluation have been strengthened with additional experiments on a 15B-MoE language model, as well as ViT and DiT architectures. Further comparisons (e.g., Peri-LN, SpanNorm) are also appreciated. The authors provided more detailed ablation studies (notably in Table 3), which help clarify the source of improvements. The last computational overhead is quantitatively handled, yet the rebuttal provides a minimally sufficient clarification; however, the AC notes that the proposed Siamese design is unlikely to introduce prohibitive overhead under modern parallelism, especially given its compatibility with residual connections.

Overall, given that the primary concerns have been reasonably addressed within the short rebuttal period, and considering the novelty of exploring a design space beyond conventional pre-norm and post-norm formulations, as well as the importance of the problem, the AC believes that the strengths of this paper outweigh its current weaknesses. Therefore, the AC recommends acceptance, subject to the authors incorporating all additional experiments and clarifications from the rebuttal into the final version of the paper.